# A Metastructure Based on Amorphous Carbon for High Efficiency and Selective Solar Absorption

**DOI:** 10.3390/nano14070580

**Published:** 2024-03-27

**Authors:** Junli Su, Gang Chen, Chong Ma, Qiuyu Zhang, Xingyu Li, Yujia Geng, Bojie Jia, Haihan Luo, Dingquan Liu

**Affiliations:** 1Shanghai Key Laboratory of Optical Coatings and Spectral Modulation, Shanghai Institute of Technical Physics, Chinese Academy of Sciences, Shanghai 200083, China; sujl@shanghaitech.edu.cn (J.S.); gangchen@mail.sitp.ac.cn (G.C.); machong@mail.sitp.ac.cn (C.M.); zhangqiuyu@mail.sitp.ac.cn (Q.Z.); lixy12@shanghaitech.edu.cn (X.L.); gengyj2022@shanghaitech.edu.cn (Y.G.); jiabojie22@mail.ucas.ac.cn (B.J.); 2School of Physical Science and Technology, ShanghaiTech University, Shanghai 200031, China; 3School of Optoelectronics, University of Chinese Academy of Sciences, Beijing 100049, China

**Keywords:** solar energy, broadband absorption, amorphous carbon, cermet nanocomposite material, quasi-resonant cavities

## Abstract

Efficient solar thermal conversion is crucial for renewable clean energy technologies such as solar thermal power generation, solar thermophotovoltaic and seawater desalination. To maximize solar energy conversion efficiency, a solar selective absorber with tailored absorption properties designed for solar applications is indispensable. In this study, we propose a broadband selective absorber based on amorphous carbon (a-C) metamaterials that achieves high absorption in the ultraviolet (UV), visible (Vis) and near-infrared (NIR) spectral ranges. Additionally, through metal doping, the optical properties of carbon matrix materials can be modulated. We introduce Ti@a-C thin film into the nanostructure to enhance light absorption across most of the solar spectrum, particularly in the NIR wavelength band, which is essential for improving energy utilization. The impressive solar absorptivity and photothermal conversion efficiency reach 97.8% and 95.6%, respectively. Notably, these superior performances are well-maintained even at large incident angles with different polarized states. These findings open new avenues for the application of a-C matrix materials, especially in fields related to solar energy harvesting.

## 1. Introduction

Solar energy stands out as a clean, plentiful, and sustainable power source, capable of not only meeting present and future global energy needs but also significantly reducing worldwide carbon dioxide emissions [1]. The escalating demand for environmentally friendly and sustainable energy solutions has spurred extensive efforts in harnessing solar energy effectively. Currently, solar energy capture predominantly relies on two approaches: solar photovoltaic strategies [2,3], which involve the conversion of photon energy directly into electricity using various photovoltaic devices, and solar thermal strategies [4,5], where solar photons are converted into heat through solar thermal systems. Among the two approaches, solar thermal technology holds particular promise due to its utilization of a broader spectrum across the entire solar range, resulting in higher conversion efficiency. Consequently, recent years have witnessed a substantial increase in the practical application of solar thermal technology, spanning areas such as solar heating [6], solar thermal power generation [7], solar thermophotovoltaic [8], and seawater desalination [9]. In solar thermal systems, solar collectors with solar selective absorption play a crucial role in system performance. The fundamental principle underlying photothermal conversion entails the collection of solar radiation energy through the meticulous design of absorption materials and structures. Subsequently, the collected energy is transformed into thermal energy through the interaction of light with matter. The efficacy of solar-selective absorbers, characterized by maximal solar absorption in the solar spectrum and the suppression of heat loss in the infrared region, directly dictates the ultimate photothermal efficiency. This underscores the significance of achieving high solar absorptivity (α) and low thermal emissivity (ε) for optimal conversion efficiency. However, a daunting challenge remains in the design and development of cost-effective, robust solar-spectrum-selective absorbers capable of concurrently exhibiting high thermal stability and solar thermal conversion efficiency.

In recent years, advancements in nanoscale fabrication technology have facilitated the development of metamaterial solar absorbers that demonstrate impeccable electromagnetic responses in specific wavelength ranges. To broaden the absorption bandwidth, a common strategy involves combining various resonators of different sizes within a unit cell to excite localized surface plasmon resonances (LSPR) or propagating surface plasmon resonances (PSPR). Metals [10,11,12], particularly noble metals like gold and silver, are the most prevalent materials for surface plasmon resonances in nanoscale particles or resonators of common periodic structures, exhibiting ideal optical responses in narrow bands of visible light [13,14]. For the last few years, carbon-based materials, such as graphene, graphene oxide, carbon nanotubes (CNT), graphite, carbon black and carbon dots, have drawn the interest of researchers due to their strong visible light absorption and the high photothermal conversion capabilities stemming from their conjugated structures (delocalized p-bonds). For example, Zhequn Huang et al. realized a ruthenium–carbon nanotube (Ru-CNT) nanocomposite refractory selective solar absorber with multiscale nanoparticle-on–nanocavity plasmonic modes, achieving a total solar absorption of 96.1% with a distinct spectral cutoff at 2.21 μm [15]. Lin, H et al. experimentally demonstrated a 90-nanometer-thick graphene metamaterial with around 85% absorptivity to unpolarized, visible and near-infrared light across nearly the entire solar spectrum (300–2500 nm) [16]. Keng-Te Lin et al. designed and prepared a 3D graphene metamaterial absorber that achieved high absorption, of over 95%, in the 280–1600 nm band [17]. It is observed that due to its distinctive p-p* interband transition processes, a-C inherently exhibits a discernible light absorption capability [18]. In addition, a-C possesses commendable mechanical strength and chemical stability, coupled with relatively high visible light absorption and infrared transmittance [19]. Consequently, it is regarded as an effective protective material.

In this paper, we introduce and numerically illustrate a novel polarization-independent, wide-angle, broadband, and highly effective selective solar thermal metamaterial absorber that essentially spans the entire solar spectrum. The absorber design is based on a-C nanocylinder structures and cermet nanometer thin-film structures. Due to the resonances of a-C LSPR, including the electric dipole (ED) and magnetic dipole (MD) modes, an optical trapping structure is formed in the a-C nanocylinders. Within the multilayer film, the integration of an ultra-broadband solar-spectrum-absorbing material, Ti@a-C cermet nanocomposite, along with silicon dioxide and thin a-C layers acting as quasi-resonant cavities, significantly enhances absorption, especially in the NIR range. Comparatively, the a-C metastructure absorber (a-C MSA) achieves outstanding performance, with a high absorption of 97.8% across the 300–2500 nm range and an impressive solar thermal conversion efficiency reaching up to 95.6% at 373.15 K. Notably, it exhibits insensitivity to polarization states and maintains a robust absorption exceeding 95% across incident angles ranging from 0 to 60°. Furthermore, the simplicity of the nanoscale structure eliminates the need for intricate manufacturing processes involving numerous, multi-sized components, thereby demonstrating the great potential of a-C matrix materials in the field of broadband absorption.

## 2. Structure and Model

Figure 1a,b illustrates the three-dimensional schematic and cross-sectional view of the unit structure of the proposed a-C MSA array, consisting of top a-C nanocylinders and a bottom quasi-resonant cavity structure. The Ti@a-C cermet thin film, serving as the ultra-broadband absorbing material, was designed and prepared using a co-sputtering method in our magnetron sputtering system with Ti and graphite targets. The a-C material and dielectric material (SiO_2_) were also fabricated into thin-film samples by magnetron sputtering deposition, and their optical parameters were obtained through ellipsometry measurements. Tungsten (W) was chosen as the metal substrate due to its high melting point, high infrared reflectance, excellent corrosion resistance, and abundance. The optical constants of W are sourced from Palki’s book [20]. The proposed solar absorber can be fabricated using vacuum film deposition equipment and electron beam lithography (EBL) [21]. As depicted in Figure 1b, the structural geometric parameters consist of the radius of a-C nanocylinder *r*, the height *h*, the thickness of the upper a-C layer *t*_1_ in the film structure, the thickness of the SiO_2_ dielectric layer *t*_2_, the thickness of the lower Ti@a-C cermet layer *t_3_* and the period *p*. To examine the absorption traits, we employed the Finite Difference Time Domain (FDTD) method to study and enhance the spectral selectivity of the proposed solar absorber [22]. In the simulations, the temperature was set to room temperature, and the impact of temperature on the optical constants of the materials was neglected. Periodic boundary conditions were implemented in the x and y directions, while perfectly matched layers (PML) were placed in the z direction. To balance simulation time and numerical accuracy, the grid size along the x, y, and z directions was set to 1 nm. Due to the thickness of the bottom W substrate (200 nm) being significantly greater than the radiation penetration depth, the transmittance T(λ) could be neglected. Therefore, the absorptance A(λ) could be indirectly acquired as A(λ) = 1 − R(λ), where R(λ) represents the reflectance.

For a solar thermal system, a selective absorber should achieve perfect absorption in UV–Vis–NIR range (0.3–2.5 μm) and emit zero radiation in the mid-infrared thermal radiation range (2.5–30 μm). The performance of the a-C MSA can be characterized by its total solar radiation absorptivity $\alpha_{total}$ and total thermal emissivity $\varepsilon_{total}$ at normal incidence, expressed as follows [23,24]:(1)$$\alpha_{total}=\frac{\int_{0.3\;\mu m}^{2.5\;\mu m}A\left(\lambda \right)I_{S}\left(\lambda \right)d\lambda }{\int_{0.3\;\mu m}^{2.5\;\mu m}I_{S}\left(\lambda \right)d\lambda }$$
(2)$$\varepsilon_{total}=\frac{\int_{2.5\;\mu m}^{30\;\mu m}\varepsilon \left(\lambda \right)I_{B}\left(\lambda ,T\right)d\lambda }{\int_{2.5\;\mu m}^{30\;\mu m}I_{B}\left(\lambda ,T\right)d\lambda }$$

According to Plank’s law, IBλ,T=2πhc02λ−5exphc0λkT−1 represents the black body radiation power density of the absorber at temperature *T*. $\varepsilon \left(\lambda \right)$ is the spectral emissivity of the absorber at this temperature. In adherence to Kirchhoff’s law of thermal radiation, the spectral emissivity of the absorber is equal to the absorptivity, i.e., $\varepsilon \left(\lambda \right)=A\left(\lambda \right)$, when the a-C MSA is in thermal equilibrium with the environment [25]. As per Stefan–Boltzmann’s law, assuming negligible thermal losses due to conduction and convection, the overall photo-thermal conversion efficiency of a-C MSA is calculated by:(3)$$\eta =\alpha_{total}-\varepsilon_{total}\frac{\sigma \left(T_{abs}^{4}-T_{air}^{4}\right)}{CI_{S}}$$
where $\sigma =5.67\times 10^{-8}\;W\cdot m^{-2}\cdot K^{-4}$ represents the Stefan–Boltzmann constant, C is the solar concentration factor, $I_{S}$ is the solar irradiance at AM 1.5 G, which is about $1000\;W\cdot m^{-2}$ [26], $T_{abs}$ and $T_{air}$ are the temperatures of the absorber and air.

## 3. Results and Discussion

### 3.1. a-C MSA for Solar Energy Harvesting

First, Ti@a-C cermet film (~100 nm), a-C film (~100 nm) and SiO_2_ film (~100 nm) were sequentially deposited on silicon wafers. The corresponding data were obtained using an ellipsometer (Appendix A); then, the optical constants were estimated by fitting different dispersion models (the Drude–Lorentz model for Ti, the Drude–Lorentz and Tauc–Lorentz models for Ti@a-C and a-C, and the Sellmeier model for SiO_2_). As depicted in Figure 1c, the refractive index of Ti@a-C cermet spans from 2.5 to 3, accompanied by an extinction coefficient concentrated within the range of 0.6–1, exhibiting minimal dispersion. This manifests a characteristic of ultra-broadband absorption, particularly demonstrating heightened absorption in the infrared spectrum of 1000–2500 nm. Notably, a-C exhibits a refractive index hovering around 2, showing discernible extinction absorption in 300–1200 nm, thus indicating its absorption capability in the Vis and NIR short wavelength region. Simultaneously, SiO_2_ maintains a consistent refractive index of approximately 1.45, coupled with an extinction absorption nearing zero.

Combining the optical properties of a-C and Ti@a-C cermet materials, the optimal structural parameters of the a-C MSA are as follows: r=120 nm, h=150 nm, $t_{1}=20\;nm$, $t_{2}=30\;nm$, $t_{3}=70\;nm$ and p=400 nm. In Figure 1d, the reflection spectrum, solar irradiance flux density under one sun concentration, and infrared radiative energy density of a-C MSA under TE polarization (electrical field in the x direction) at normal incidence are presented. The blue reflectance curve indicates the excellent spectral selectivity of the proposed solar absorber. The absorber achieves nearly perfect absorption in the UV–Vis–NIR spectra, with a spectral reflectance below 2% in the 300–1500 nm range, and also below 10% in the 300–1700 nm range. In the infrared band beyond 3 mm, the spectral reflectance persists above 95%, indicating a thermal emissivity below 5%. As demonstrated in Figure 1a,b, the a-C MSA exhibits central rotational symmetry along the z axis, with no differentiation in absorption between TE- and TM-polarized light under normal incidence. In this work, TE-polarized light is chosen as the incident light source.

The broadband absorption capabilities of the a-C MSA are evident in Figure 2a, where the purple spectral absorptance curve indicates nearly perfect coverage of the majority of the solar radiation spectrum. It maintains a spectral absorptance above 97% in the 350–1500 nm range, with only a minimal amount of residual energy unabsorbed below 350 nm in the UV range and the NIR range of 2000–2500 nm. For the subwavelength- and wavelength-sized metastructure absorber, which exhibits sensitive responses to both electric and magnetic fields, the optical properties cannot be adequately described solely by the optical parameters of a-C and Ti@a-C materials. To gain a deeper understanding of the a-C MSA’s broadband high-absorption characteristics, its overall structure was treated as an equivalent medium. Impedance matching theory [27] was applied to calculate the equivalent impedance of the a-C MSA in the 300–2500 nm range, as depicted in Figure 2b. The variations of the real and imaginary parts of the equivalent impedance were closely monitored. When the a-C MSA achieves impedance matching with free space, nearly perfect absorption can be achieved. The real part of the impedance of free space is defined as $Re\left(Z_{free\;space}\right)=\sqrt{\mu \left(\lambda \right)/\varepsilon \left(\lambda \right)}=1$, and the imaginary part is $Im\left(Z_{free\;space}\right)=0$, which are represented by orange and brown dashed lines, respectively. While the real part of the effective impedance of the absorber is close to 1 ($Re\left(Z_{eff}\right)\approx 1$), and the imaginary part is close to 0 ($Im\left(Z_{eff}\right)\approx 0$), the absorber exhibits high absorption. Within the wavelength range of 300–1500 nm, the real and imaginary parts of the equivalent impedance closely follow the orange and brown dashed lines, respectively, indicating high absorption of over 97% in the absorptance spectrum. Beyond 1700 nm, the real and imaginary parts of the equivalent impedance undergo drastic changes, deviating from 1 and 0, resulting in a significant increase in reflection in the spectrum curve. This overall behavior attests to the selective absorption characteristics of the superstructure absorber.

### 3.2. Underlying Mechanisms of the Broadband Absorption

To further reveal underlying mechanisms of the broadband absorption in our proposed solar-selective absorber, we analyzed the electric and magnetic field distributions in the x-z plane at five different wavelengths. As depicted in Figure 3a, the electric field distribution reveals the formation of localized surface plasmon resonance (LSPR) within the a-C nanocylinder at wavelengths of 310, 465, 750, 1000, and 1350 nm. The resonance modes are dominated by the electric dipole mode (ED), with a distinctive shift in the resonance position on the a-C nanocylinders as the wavelength increases. In Figure 3b, at 310 nm, the magnetic quadrupole resonance mode (MQ) is observed within the a-C nanocylinder, while a propagating surface plasmon resonance mode (PSPR) arises at the Ti@a-C cermet and SiO_2_ interface, owing to the inherent metallic properties of the Ti@a-C nanocomposite. At 465 nm and 710 nm, a mutual coupling of MQ and MD occurs within the a-C nanocylinders, gradually transitioning into the predominant MD mode. Within the foundational thin-film structure, the PSPR can be observed at the interface between the Ti@a-C cermet layer and SiO_2_ layer. This phenomenon is denoted as quasi-resonant cavity resonance absorption. Moreover, a PSPR mode is also observed at the interface between Ti@a-C cermet layer and metallic W substrate. This occurrence is attributed to the dielectric characteristics inherent in the Ti@a-C cermet nanocomposite. At 1000 nm, the absorption mainly involves the interactions between MD and ED modes, along with the interaction of PSPR modes at the interfaces of the a-C and SiO_2_ layers. The PSPR mode at the interface between the Ti@a-C cermet and W substrate predominates. At 1350 nm, the magnetic field is predominantly distributed in the Ti@a-C layer and at the interfaces of the Ti@a-C layer, owing to interband transitions in the Ti nanoparticles within the Ti@a-C cermet, the plasmon resonance of embedded Ti nanoparticles in the amorphous carbon matrix [28], and the absorption of PSPR at the upper and lower surfaces of the Ti@a-C layer [29]. Notably, when the resonant wavelengths of MD and ED modes are close to each other, the interaction of MD and ED modes results in a Huygens’ metasurface, leading to the suppression of backward scattering [30,31]. This signifies a reduction in reflection within the 700–1500 nm range, a phenomenon referred to as the Kerker condition [32]. In summary, the intrinsic absorption of a-C and Ti@a-C materials, coupled with ED, MD, and MQ resonances within a-C nanocylinders, as well as the PSPR modes at various interfaces within the Ti@a-C quasi-resonant cavity, collectively contribute to the broadband absorption characteristics of the metastructure.

### 3.3. Geometric Effects on Spectral Absorption Performance at Normal Incidence

Based on the aforementioned absorption mechanisms, we conducted an investigation into the impact of the geometric dimensions of the designed a-C MSA on its absorption performance.

As illustrated in Figure 4a, an increase in the radius of a-C nanocylinders leads to a gradual enhancement of absorption within the 500–1000 nm wavelength range. This phenomenon is attributed to the escalating interaction between ED and MD modes. However, excessively large nanocylinder radii (140 nm and 160 nm) hinder light penetration into the underlying Ti@a-C resonant cavity structure, resulting in a weakened absorption beyond 800 nm in the NIR range. In Figure 4b, when the nanocylinder height approximates its diameter, it not only facilitates the excitation of the MQ mode but also fosters the coupling of ED and MD modes, leading to enhanced absorption at around 350 nm and 750 nm.

In the Ti@a-C resonant cavity structure, as illustrated in Figure 5a,b, the thickness of the a-C and SiO_2_ exhibits a relatively minor influence on absorption. Particularly in the UV range around 350 nm, thin layers of a-C and SiO_2_ prove advantageous for exciting the PSPR at the Ti@a-C and SiO_2_ interface. The thickness of a-C can effectively regulate the cut-off wavelength [$R\left(\lambda_{cut\text{-}off\;wavelength}\right)=0.5$] of the metastructure absorber. As depicted in Figure 5c, notably, the optical response in the infrared range is particularly pronounced, with the thickness of the Ti@a-C layer adeptly modulating the wide-spectrum absorption beyond 700 nm in the near-infrared. This modulation is attributed to the inherent broadband absorption characteristics within the Ti@a-C cermet layer, along with the PSPR absorption between the upper and lower surfaces of the Ti@a-C cermet layer. However, excessively thick Ti@a-C cermet layers impede the excitation of PSPR at the interface between the metal substrate and Ti@a-C cermet layer, resulting in reduced absorption in the 600–1800 nm range. Moreover, with an increasing thickness of Ti@a-C, the cut-off wavelength of the absorber undergoes a redshift, allowing for precise control over the energy response range of the absorber.

The change in periodicity fundamentally modifies the distance between adjacent a-C nanocylinders, thereby controlling the ED response between neighboring structures and influencing the incident light flux into the underlying resonant cavity structure. As illustrated in Figure 6a, within the range of structural periodicity variations from 300 to 550 nm, an increase in *p* leads to the enhancement and subsequent attenuation of absorption in the visible light spectrum, reaching an optimum value at 400 nm. Comparing the absorption spectra of various absorber structures, as shown in Figure 6b, it is evident that tungsten exhibits high absorption in the UV–Vis–NIR ranges due to its elevated imaginary part of refractive index. The responses of ED, MD, and MQ in the a-C nanocylinder structure significantly enhance absorption in the UV and Vis region. The remarkable broadband characteristics of the Ti@a-C cermet material, coupled with the resonant effects of the quasi-resonant cavity, substantially elevate its absorption across the entire solar radiation spectrum, particularly in the NIR range. The integration of a dual-absorber structure results in an almost perfect optical response in the designed a-C MSA.

### 3.4. The Effect of Polarization Angle and Oblique Incident Angle

To obtain as much solar radiation energy as possible, the proposed absorber should have a structure that can eliminate dependence on the polarization state of incident light. The numerical results outlined above demonstrate that the proposed metamaterial absorber possesses superior bandwidth tunability for selective solar energy absorption, making it a promising candidate for numerous practical applications. Absorptance tests were conducted on the a-C MSA within the range of incident polarization angles from 0 to 90°, as illustrated in Figure 7. Due to the central symmetry of the periodic structure in the metasurface, the absorptance remains nearly unchanged with variations in the incident polarization angle.

Furthermore, to maximize energy-harvesting and utilization efficiency, wide-angle absorption plays a crucial role in solar selective absorbers. Here, we investigated the angular dependence of the proposed solar selective absorber on the absorptance for both TM and TE polarizations, as depicted in Figure 8a,b. The computational results indicate that, for both TM and TE modes, the absorption bandwidth and peak position exhibit minimal variations within the range of incident angles from 0 to 60°. The absorption for unpolarized light can be calculated using the average of the TE and TM absorptions ($\alpha \left(Unpolarized\right)=\alpha \left(TE\right)/2+\alpha \left(TM\right)/2$), as depicted in Figure 8c, which demonstrates that the absorber maintains over 95% high absorptance within the incident angle range of 0–60°. This underscores the robustness and effectiveness of the proposed a-C MSA, suggesting its potential for versatile applications in energy harvesting and utilization.

Assuming the absorber temperature is 100 °C and the ambient temperature is 26 °C, the calculated solution using Formulas (1)–(3) reveals that, under normal incidences, the total solar radiation absorptivity ($\alpha_{total}$) of the a-C MSA is 97.8%, and the total thermal emissivity ($\varepsilon_{total}$) above 2.5 μm is only 2.2%. Consequently, the overall solar thermal conversion efficiency (η) reaches an impressive 95.6%. Based on the above analysis, the designed metamaterial absorber exhibits outstanding optical absorption characteristics. This feature makes it highly applicable in various energy conversion domains, capitalizing on its efficient solar selective absorption properties to optimize solar energy utilization.

## 4. Conclusions

In summary, we propose a metamaterial nanostructure absorber based on a-C nanocylinders and Ti@a-C cermet nanocomposite, contributing to the repertoire of materials with plasmonic electromagnetic responses. The calculated absorption/emission reveal that, within the solar spectrum range of 300 to 1500 nm, the absorptivity > 95%, while in the mid-infrared range, the emissivity < 5%, demonstrating excellent spectral selectivity. Particularly noteworthy is the absorber’s theoretical photo-thermal conversion efficiency, reaching up to 95.6% at 373.15 K, showcasing its potential for high-performance solar thermal applications. Simulation results suggest that incorporating a-C nanopatterns into the Ti@a-C cermet resonant cavity structure can further enhance light–matter interactions, achieving near-perfect solar absorption in the UV–Vis–NIR spectra. Furthermore, the absorption feature can be easily regulated by slightly adjusting structural parameters, making the absorber practically appealing. Due to the high symmetry, the a-C solar absorber maintains excellent solar selectivity for both TM and TE polarizations over a wide-angle range, indicating its omnidirectional and polarization independence. These advantages, including spectral selectivity, high efficiency, and wide-angle absorption, benefit a-C MSA for diverse applications in solar energy utilization.

## Figures and Tables

**Figure 1 nanomaterials-14-00580-f001:**
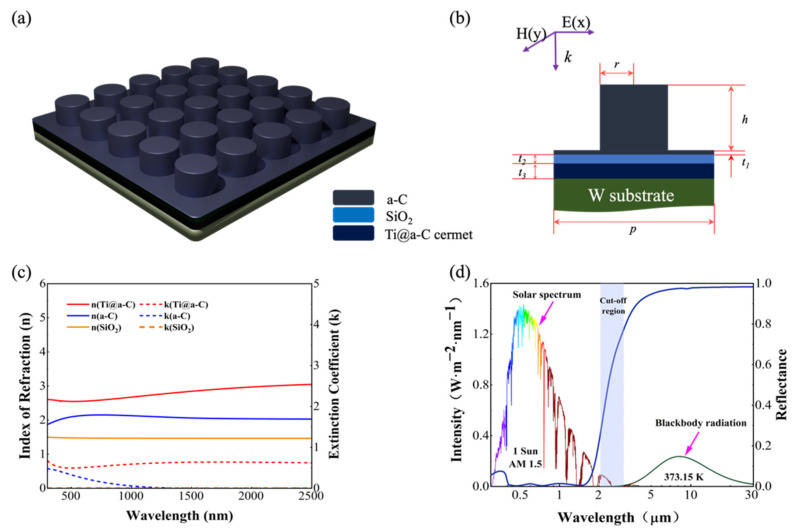
(**a**) 3D schematic of the a-C MSA; (**b**) 2D cross-section schematic of the unit cell; (**c**) optical constants of Ti@a-C, a-C and SiO_2_: refractive index n and extinction coefficient k; and (**d**) solar spectrum (AM 1.5), blackbody radiation spectrum at 100 °C interspersed with reflection spectrum of the sample, and cut-off region.

**Figure 2 nanomaterials-14-00580-f002:**
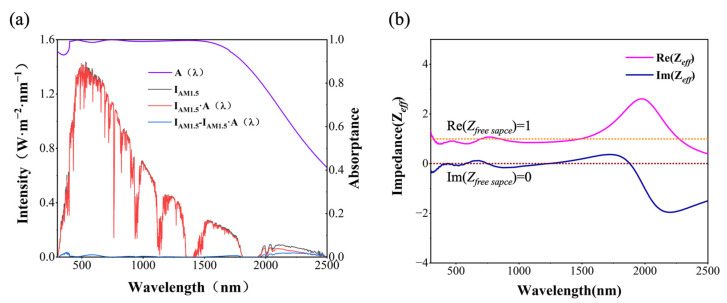
(**a**) Solar spectral (AM 1.5), spectrum absorptance of the proposed a-C MSA, and residual solar irradiance in the UV–Vis–NIR regions; and (**b**) the calculated real and imaginary parts of the effective impedance for the proposed a-C MSA.

**Figure 3 nanomaterials-14-00580-f003:**
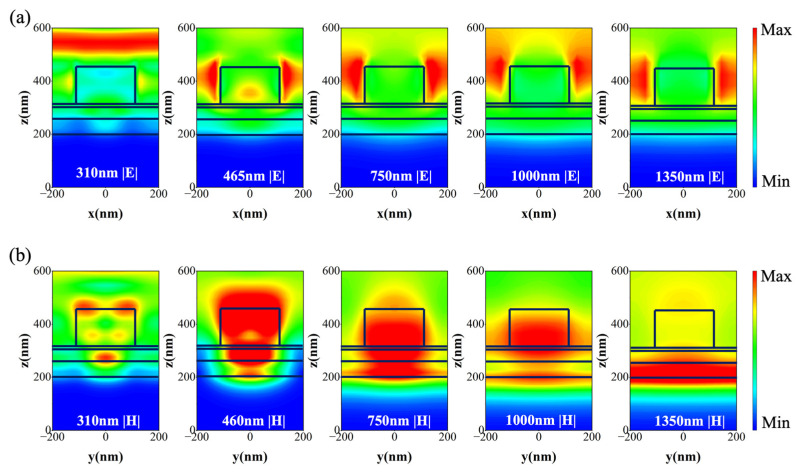
Distributions of the normalized electric field (|E|) and magnetic field (|H|) in the x-z and y-z plane at wavelengths of 310 nm, 465 nm, 750 nm, 1000 nm, and 1350 nm, respectively: (**a**) electric field in the x-z plane; and (**b**) magnetic field in the y-z plane.

**Figure 4 nanomaterials-14-00580-f004:**
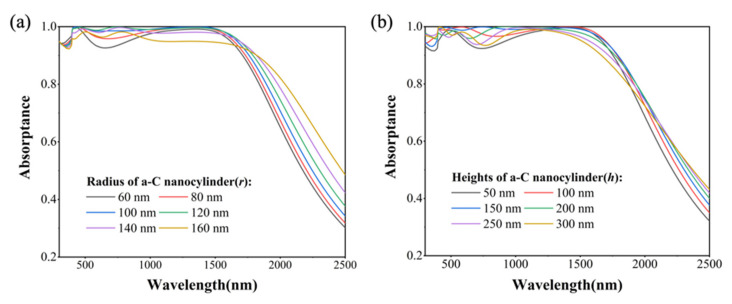
Energy dissipation within the a-C nanocylinder: (**a**) the contributions of the radius (r); and (**b**) the contributions of the height (h).

**Figure 5 nanomaterials-14-00580-f005:**
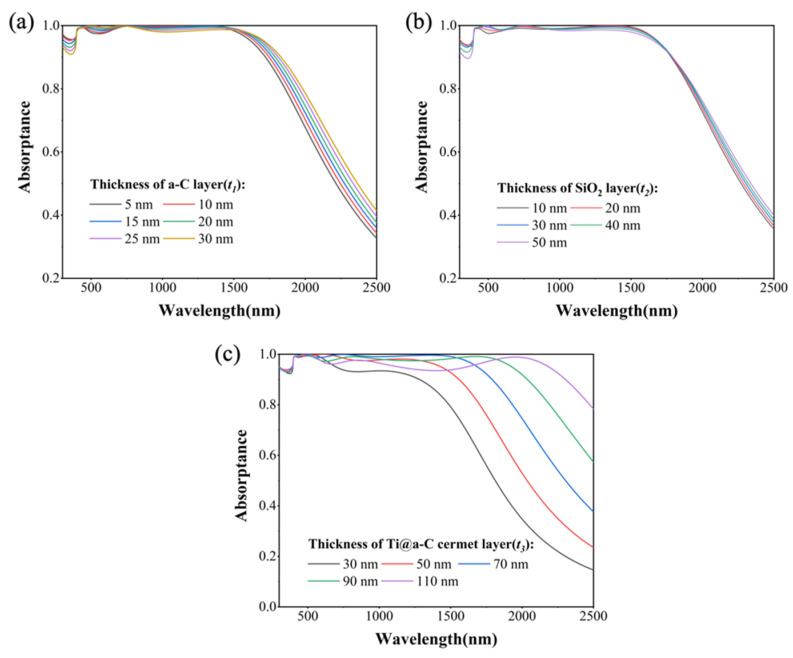
(**a**) Absorption spectra of absorbers with the thickness (*t*_1_) of a-C layer varies from 5 to 30 nm; (**b**) absorption spectra of absorbers with the thickness (*t*_2_) of the intermediate SiO_2_ layer varies from 10 to 50 nm; and (**c**) absorption spectra of absorbers with the thickness (*t*_3_) of the Ti@a-C cermet layer varies from 30 to 110 nm.

**Figure 6 nanomaterials-14-00580-f006:**
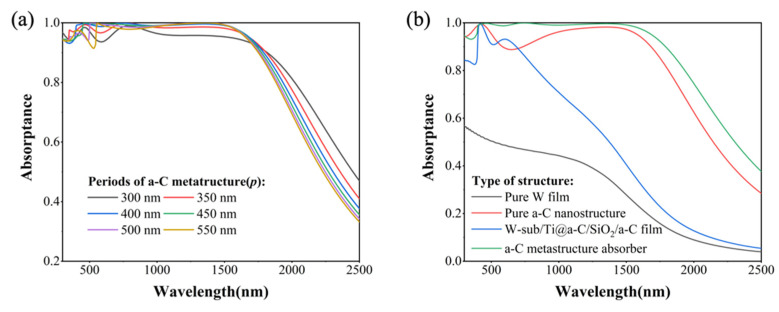
(**a**) The effect of periods (*p*) on the absorptance of the a-C meta-structure absorber; and (**b**) absorption spectra comparison of the simple W film, Ti@a-C/SiO_2_/a-C film, pure a-C nanostructure absorber and a-C metastructure absorber.

**Figure 7 nanomaterials-14-00580-f007:**
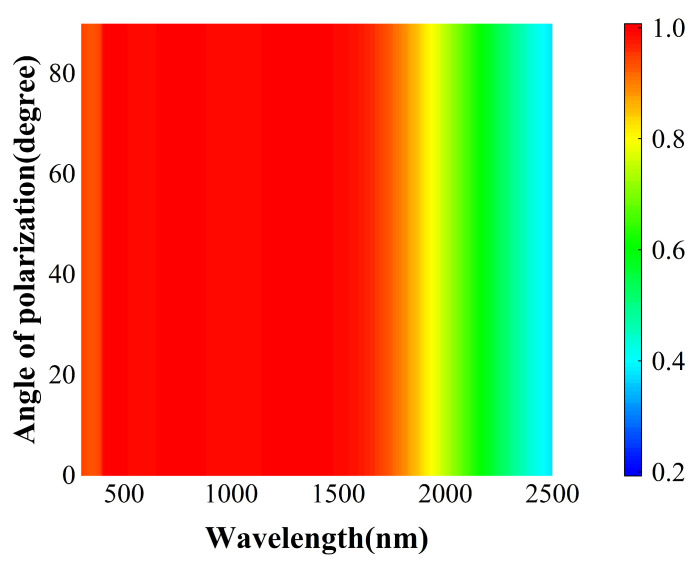
Absorption spectra of the present absorbers with different angles of polarization at normal incidence.

**Figure 8 nanomaterials-14-00580-f008:**
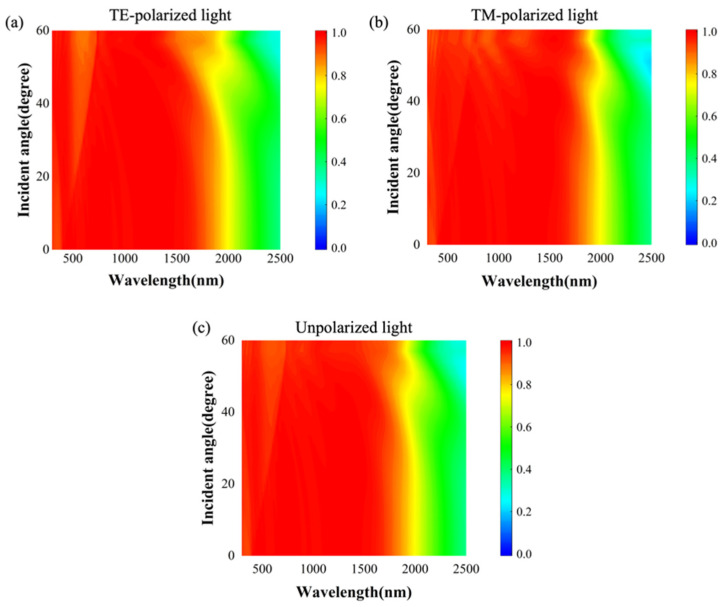
Absorption spectra of absorbers with different incident angles: (**a**) TM polarized wave; (**b**) TE polarized wave; and (**c**) unpolarized light.

## Data Availability

Data are contained within the article and Appendix A.

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
