# Peer review of "A Metastructure Based on Amorphous Carbon for High Efficiency and Selective Solar Absorption"

_nanomaterials, 2024, doi:10.3390/nano14070580_

Round 1

Reviewer 1 Report

Comments and Suggestions for Authors

In this work, the authors introduce a broadband selective absorber utilizing amorphous carbon (a-C) metamaterials, achieving nearly perfect absorption across the ultraviolet (UV), visible (Vis), and near-infrared (NIR) spectral ranges. The performance of the proposed structure is observed for different polarizations and several angles of incidence. The shown results are interesting for solar energy harvesting. The article is well-written and the results are of interest. I support the article for its publication after the following questions are addressed:

1)To understand the origin of the modes giving rise to the absorption, it would be recommended to carry out a multipolar decomposition analysis.

2)It is mentioned that the interferential effects between dipolar electric and dipolar magnetic modes give rise to directionality properties (Huygens’ metasurface). Which is the role of the higher multipolar orders? In particular, which is the influence of the magnetic quadrupolar mode in the directionality?

3)In some previous works, the interferential effects between electric and magnetic dipolar resonances have been numerically and experimentally proposed to increase the efficiency of solar cells in broadband spectral regions. However, these works have not been cited in the article. I recommend the authors to consider these references in the introduction of the manuscript.

[1] Barreda, Á.I., Saleh, H., Litman, A. et al. On the scattering directionality of a dielectric particle dimer of High Refractive Index. Sci Rep 8, 7976 (2018). https://doi.org/10.1038/s41598-018-26359-8
[2] J. van de Groep and A. Polman, "Designing dielectric resonators on substrates: Combining magnetic and electric resonances," Opt. Express 21, 26285-26302 (2013)

Reviewer 2 Report

Comments and Suggestions for Authors

The authors have presented a study on a polarization-insensitive UV-Vis-NIR metamaterial-based absorber composed of a-C nanocylinder structures and nanometer-thin film structures. They have demonstrated that their designed absorber maintains robust absorption levels exceeding 95% across incident angles ranging from 0 to 60°. Additionally, they have shown a high absorption rate of 97.8% within the 300-2500 nm range and a solar thermal conversion efficiency of up to 95.6% at 373.15 K. The proposed design of the absorber is intriguing, and the results are promising. However, there are several areas that require attention:

1.       The manuscript lacks a comprehensive literature review. Including a comparison table with key performance indicators (e.g., absorption, bandwidth, efficiency, polarization dependence) could help highlight the potential of the proposed design.

2.       The authors mention that the thickness of the Ti@a-C cermet layer (t3) predominantly controls NIR absorption and mention reduced absorption in the 600-1800 nm range. However, for wavelengths above 1800 nm, absorption is significantly reduced for the chosen optimal thickness value (t3=70 nm). This issue should be clearly explained in the manuscript. Consideration of higher thickness values may potentially improve overall absorption.

3.       Please remove all hyperbolic sentences that are clearly inappropriate for a scientific manuscript.

4.       The English language in the manuscript requires significant improvement, as there are many grammatical errors, typos, and missed words. This can be addressed by having a native English-speaking colleague revise the manuscript or by utilizing professional editing services, such as those offered by Elsevier's Language Editing Services (https://webshop.elsevier.com/language-editing-services/language-editing/).

Comments on the Quality of English Language

The English language in the manuscript requires significant improvement, as there are many grammatical errors, typos, and missed words. This can be addressed by having a native English-speaking colleague revise the manuscript or by utilizing professional editing services, such as those offered by Elsevier's Language Editing Services (https://webshop.elsevier.com/language-editing-services/language-editing/).

Round 2

Reviewer 1 Report

Comments and Suggestions for Authors

I support the article for its publication 

Reviewer 2 Report

Comments and Suggestions for Authors

I agree with authors corrections and responses